# Treating Pulmonary Fibrosis with Non-Viral Gene Therapy: From Bench to Bedside

**DOI:** 10.3390/pharmaceutics14040813

**Published:** 2022-04-07

**Authors:** Teng Huang, Jia Gao, Long Cai, Hao Xie, Yuhan Wang, Yi Wang, Qing Zhou

**Affiliations:** 1National Health Commission Key Laboratory of Pulmonary Diseases, Department of Respiratory and Critical Care Medicine, The Center for Biomedical Research, Tongji Hospital, Tongji Medical College, Huazhong University of Science and Technology, Wuhan 430074, China; teahuangteng@126.com (T.H.); d202081861@hust.edu.cn (J.G.); d201981701@hust.edu.cn (H.X.); d202181917@hust.edu.cn (Y.W.); 2Unit 95969 of the People’s Liberation Army, Wuhan 430000, China; clementtaylor@gmail.com

**Keywords:** idiopathic pulmonary fibrosis, gene therapy, non-viral delivery systems, nanoparticles

## Abstract

Idiopathic pulmonary fibrosis (IPF) is a chronic, progressive lung disease characterized by irreversible lung scarring, which achieves almost 80% five-year mortality rate. Undeniably, commercially available pharmaceuticals, such as pirfenidone and nintedanib, exhibit certain effects on improving the well-being of IPF patients, but the stubbornly high mortality still indicates a great urgency of developing superior therapeutics against this devastating disease. As an emerging strategy, gene therapy brings hope for the treatment of IPF by precisely regulating the expression of specific genes. However, traditional administration approaches based on viruses severely restrict the clinical application of gene therapy. Nowadays, non-viral vectors are raised as potential strategies for in vivo gene delivery, attributed to their low immunogenicity and excellent biocompatibility. Herein, we highlight a variety of non-viral vectors, such as liposomes, polymers, and proteins/peptides, which are employed in the treatment of IPF. By respectively clarifying the strengths and weaknesses of the above candidates, we would like to summarize the requisite features of vectors for PF gene therapy and provide novel perspectives on design-decisions of the subsequent vectors, hoping to accelerate the bench-to-bedside pace of non-viral gene therapy for IPF in clinical setting.

## 1. Introduction

Idiopathic pulmonary fibrosis (IPF) is a chronic, progressive lung disease, of which the pathogenesis is still masked [1,2]. The main clinical manifestations of patients with IPF are unexplained exertional dyspnea, chronic dry cough, and velcro-like crackles. The prevalence of IPF appears to be increasing and the incidence of the disease is higher in Europe and North America (3 to 9 cases per 100,000 person-years) than in East Asia and South America (fewer than 4 cases per 100,000 person-years) [2]. IPF usually occurs in older adults (>50 years), and men are at higher risk than women [3]. In addition to factors such as aging, smoking, and dusty living environments, IPF sometimes runs in families, and several genes are involved in its development [3]. Due to the lack of effective treatments, the average life expectancy of the patients after diagnosis is usually 2.5–3.5 years [4,5,6].

Current treatments for IPF are mainly composed of nonpharmacological managements (including smoking cessation, supplemental oxygen, pulmonary rehabilitation, lung transplantation) and pharmacotherapeutic approaches [2]. Up to now, only nintedanib and pirfenidone are recommended as the first-line therapeutic drugs for the treatment of IPF [1,2]. Although these two drugs can attenuate the development of pulmonary fibrosis to a certain extent, the restricted curative effects do not reduce the five-year mortality rate of IPF patients, actually. Moreover, the accompanying side effects, such as bleeding, diarrhea, and liver toxicity, even aggravate the patients’ suffering [7,8,9,10]. In view of the limitations of current pharmacotherapeutics for IPF, safer and more effective drugs are urgently needed.

The increasing number of approved nucleic acid-based therapies has demonstrated the potential of gene therapy for various diseases, including IPF [11]. By transfecting the target cells with specific exogenous nucleotide fragment, such as DNA, mRNA, small interfering RNA (siRNA), microRNA (miRNA) mimics, and short hairpin RNA (shRNA), gene therapy is able to precisely regulate gene expression and subsequently modulate the function of target cells [12]. Lipid-based, polymeric nanoparticles, and protein/peptides are the main categories of non-viral gene delivery systems. To deliver gene cargos to the target cells with high efficiency, the employed vectors need to be well-designed, which should simultaneously possess favorable stability, desirable target affinity, optimal loading capacity, and satisfactory biocompatibility [13]. In order to achieve above goals in IPF treatment, herein we review the pathophysiological process of pulmonary fibrosis and summary gene-therapy strategies applied in treating IPF, especially the applications of non-viral vectors, hoping to accelerate the bench-to-bedside pace of non-viral gene therapy for IPF.

## 2. Pathophysiology of IPF

Idiopathic pulmonary fibrosis has been considered as the result of multiple environmental and interacting genetic risk factors. Senescence [14], tobacco smoking [15,16], air pollution [17,18], gastroesophageal reflux [19], obstructive sleep apnea [20], herpesvirus infection [21], and certain occupational exposures [22] have been identified as nongenetic risk factors for IPF. Variations in genes that are associated with host defense (*MUC5B*, *ATP11A*, *TOLLIP*) [23,24,25,26,27], telomerase length maintenance (*TERT*, *TERC*, *PARN*, *RTEL1*, *OBFC1*, *DKC1*, *TINF2*) [23,25,28,29,30], surfactant dysfunction (*SFTPC*, *SFTPA2*, *ABCA3*), and cell–cell adhesion (*DSP*, *DPP9*) increase susceptibility to IPF [31,32,33,34].

Fibrogenesis is often defined as an out-of-control repair of damaged tissues with an excessive accumulation of extracellular matrix (ECM), which consists of a clotting/coagulation phase, an inflammatory phase, a fibroblast migration/proliferation phase, and a remodeling phase [35]. Epithelial cells and endothelial cells release inflammatory mediators when tissue damage, causing an antifibrinolytic-coagulation cascade and a recruitment of inflammatory cells. Activated macrophages and neutrophils produce a variety of cytokines and chemokines that amplify the inflammatory response and trigger fibroblasts proliferation and recruitment. Fibroblasts are activated and transitioned to matrix-producing myofibroblasts, which secrete abundant ECM components (e.g., hyaluronic acid, fibronectin, proteoglycans, and interstitial collagens) and remodel lung interstitium [1,35] (Figure 1).

Transforming growth factor-β (TGF-β), fibroblast growth factor (FGF), Wnt, hedgehog, and Notch signaling pathways have been proven to be involved in the pathogenesis of pulmonary fibrosis by orchestrating the functions of effector cells (e.g., epithelial cells, macrophages, fibroblasts, and myofibroblasts) [36]. Current and potential treatments mainly concentrate on targeting these signaling pathways. For example, nintedanib and pirfenidone are two kinds of classical small-molecule drugs which are recommended for the treatment of IPF [1,2]. Nintedanib is a tyrosine kinase inhibitor targeting soluble vascular endothelial growth factor receptor (VEGFR)/fibroblast growth factor receptor (FGFR)/platelet derived growth factor receptor (PDGFR) which exhibits anti-fibrotic and anti-inflammatory effects in IPF treatment [9]. Pirfenidone plays an anti-inflammatory and anti-fibrotic effect by down-regulating TGF-β and tumor necrosis factor-α (TNF-α) pathways, which could reduce fibroblast proliferation and inhibit collagen synthesis [7,8]. Although these drugs can attenuate the development of pulmonary fibrosis to some degree, the non-specificity of the drugs’ targets may cause serious side effects, such as bleeding, diarrhea, and liver toxicity [7,8,9,10]. A study used spermidine (Spd)-modified poly(lactic-co-glycolic acid) (PLGA) NPs to encapsulate fluorofenidone (AKF) and improve the antifibrotic efficacy of the drug in the lung. After intravenous injection of the nanomedicine into paraquat-induced mice, in vivo fluorescence imaging and HPLC analysis of the distribution of the NPs in various tissues showed that the NPs were aggregated in the lungs. Histopathological analysis exhibited that fibrohyperplasia, alveolar collapse, neutrophil infiltration, and pulmonary septal injury were significantly reduced after SPD-AKF-PLGA NP treatment [37]. Another study used chitosan–sodium alginate as a vector to delivery pirfenidone to mice through transdermal administration. Skin penetration was significantly increased, with nanoparticle-loaded pirfenidone compared with pirfenidone solution [38]. Although drugs loaded by the nanoparticles can improve their distribution and reduce their toxic side effects, the fact that small molecule inhibitors have multiple targets with low specificity still limits their clinical applications. Therefore, superior pharmacotherapeutics for IPF are urgently needed.

## 3. Gene Therapeutics for Pulmonary Diseases

With the in-depth understanding of diseases and the development of gene technology, gene-based therapy has become a reality. Compared with conventional drugs that usually target proteins, genetic drugs achieve highly specific and durable therapeutic effects by introducing exogenous nucleic acids into cells to counteract the effects of defective genes. Since exogenous nucleic acids are immunogenic, intolerant to the nucleases, and are difficult to be transfected into cells, appropriate vectors are required for gene therapy [11]. These vectors use biological and chemical modifications to protect nucleic acids from degradation during the circulation, deliver them to the target tissue, and ensure the effective transfection of nucleic acids into cells. According to their biochemical characteristics, the vehicles can be classified as viral vectors and non-viral vectors. The outstanding transfection ability makes viral vectors attractive for gene therapy [39]. Although retrovirus, adenovirus, adeno-associated virus (AAV), and herpes simplex virus have been employed in clinical applications, the risks caused by their immunogenicity, carcinogenesis, and non-targeted delivery still impede the extensive utilization [12]. In addition, the limited DNA packaging capacity and high production cost of viral vectors also slow down the bench-to-bedside pace of virus-based gene therapy [12]. Compared to viral vectors, non-viral vectors exhibit lower immunogenicity, higher loading efficiency, and are easier to be synthesized, which are more tailor-made for clinical requirements and might bring new hope for the gene therapy of pulmonary fibrosis [12,40]. 

For the gene therapy of pulmonary diseases, multiple administration routes are available, such as intravenous administration and airway administration. Due to the large surface area, high membrane permeability, extensive blood vessels, and low degrading enzyme activity in lung, drugs could pass through the air–blood barrier along with gas exchange [41]. Inhalation delivery strategies, including intratracheal/intranasal instillation and nebulization, are the simplest and most common non-invasive routes for drugs to be directly infused into lungs [41]. Considering pulmonary fibrosis-pathogenic molecules (such as TGF-β) might participate in the maintenance of normal physiological functions in other organs, systemic administration of drugs would cause severe side effects. Thus, for the treatment of pulmonary diseases, the lung in situ administration through inhalation is preferable due to the attenuated side effects and enhanced curative effects, as well as reduced administration dosages [42]. 

However, there are still some obstacles that hinder gene therapeutics’ application in the treatment of pulmonary diseases. Firstly, airway epithelium is covered with negatively charged mucus which is in charge of the capture of inhaled extraneous matters. Gene carriers are also likely fixed in the mucus through polyvalent adhesion interactions. Subsequently, the presence of proteins, lipids, surfactants, ions, and various soluble macromolecules in airway mucus may impair the stability of gene vectors [41]. For example, cationic polymers and lipid-based non-viral gene vectors can easily adhere to negatively charged mucus components by electrostatic adsorption, leading to massive aggregation. Negatively charged soluble substances in the mucus may also damage non-viral gene vectors and destabilize the complexes of DNA and cationic vectors. Secondly, gene vectors in the mucus gel will be removed from the lungs by mucociliary clearance (MCC), which prevents them from reaching potential target cells efficiently [43]. Even after the gene vectors pass through the mucus layer and reach to the alveolar sac region, the negatively charged surfactants in the alveolar sac, composed of phospholipids, cholesterol, and surfactant proteins, would prevent vectors from deeply penetrating into the focus [44]. Furthermore, although alveolar macrophages seem not to be a significant barrier for gene vectors; since gene vectors are generally smaller than the size phagocytosed by macrophages [45], aggregation of gene vectors in the presence of pulmonary surfactants may make them more easily be cleared by alveolar macrophages. Moreover, there is an embarrassing dilemma in the design of vectors. If the gene vectors can overcome the above extracellular barriers, such as escaping from the phagocytosis of macrophage, the characteristics that assist vectors extracellularly would in turn prevent gene vectors from entering target cells due to inefficient endocytosis through the membrane and tight connections between cells [46] (Figure 2). Besides the above extracellular barriers, lots of intracellular barriers also impact the efficacy of gene transfection. Once internalized by the target cell, the gene vectors must pass through various intracellular barriers including, but not limited to, acidic vesicles (e.g., endosomes and lysosomes), molecules in cytoplasm, and nuclear membranes to achieve its therapeutic effect [47]. Accordingly, to conquer the above obstacles, a variety of non-viral vectors have been raised for the gene therapy of IPF, including liposomes, polymers, and proteins/peptides. We will summarize the features of each vector in the following part.

## 4. Non-Viral Gene Therapeutics

Non-viral gene therapeutics can mediate gene transfection by utilizing the physicochemical properties of non-viral vectors, which are usually formed by a polyvalent electrostatic interaction between positively charged carrier materials and negatively charged nucleic acids. Compared with viral vectors, non-viral vectors can encapsulate more nucleic acids [12]. The preparation of non-viral vectors is simple and the production cost is much cheaper than that of viral vectors [13]. In addition, non-viral vectors possess lower immunogenicity than that of viral vectors [12]. Advances in materials gift non-viral vectors with the ability to penetrate extracellular barriers, target specific cells, and enhance intracellular delivery. Lipid-based, polymeric nanoparticles, and protein/peptides are the main categories of non-viral gene delivery systems (Figure 3). Herein, we will discuss the applications of these three gene therapeutic delivery systems in the treatment of pulmonary fibrosis (Table 1). 

### 4.1. Lipid-Based Gene Vectors

Lipid-based vectors have been widely employed for the delivery of various therapeutics, such as chemotherapeutics, peptides, and proteins [65]. Nowadays, the applications of lipid-based vectors in gene therapy have been attractive, and previous studies have demonstrated that airway administration or intravenous administration of lipid-based vectors, of which the classical example is liposomes, can efficiently deliver nucleic acid to the focus of lung diseases in mouse model [42,66]. 

Liposomes are lipid-based vesicular vectors with sizes ranging from 25 nm to several microns, which are the only nanoparticles (NPs) approved by the FDA for inhalation [67]. To package the negative-charged nucleic acids, liposomes tend to be positively charged, called cationic liposomes (Figure 4A), which are usually composed of cationic lipids, neutral phospholipids, cholesterols, and polyethylene glycol (PEG). Cationic lipids, such as N-(1-(2,3-dioleoyloxy)propyl) N,N,N-triethylammonium chloride (DOTMA), 2,3-dioleyloxy-N-(2-(sperminecarboxamido)ethyl)-N,N-dimethyl-1-propanaminium (DOSPA), 1,2-dioleoyl-3-trimethylammonium-propane (DOTAP), 1,2-dimyristyloxy-propyl-3-dimethylhydroxyethylammonium bromide (DMRIE), and 3β-(N-(N’,N’-dimethylethylenediamine)-carbamoyl) cholesterol (DC-cholesterol) have hydrophobic tail groups and positively charged head groups that can condense the negatively charged nucleic acids [68]. In addition, the neutral phospholipids form a cell-membrane-like bilayer structure in which cholesterols are inserted and in charge for regulating the fluidity of the membrane. Lastly, the surface of this spherical lipoplex is used to be decorated with PEG in order to enhance the stability, reduce the nonspecific uptake, and diminish the immunogenicity. 

To overexpress some specific genes, plasmid or mRNA can be encapsulated into cationic liposomes. Epperly et al. [48] established a magnesium superoxide dismutase (MnSOD) plasmid-loaded liposome composed of 1:1 DOTMA and DOPE, which can effectively improve the irradiation-induced pulmonary fibrosis and increase the survival rate of PF-bearing mice by enhancing SOD2 expression. Another common application of gene therapy is to silence the expression of target genes. Wang et al. [50] reported that the prevention of secreted protein acidic and cysteine rich (SPARC) expression by *SPARC*-specific siRNA encapsulated in DharmaFECT™ 1, which is a cationic lipid compound, could downregulate collagen expression and subsequently attenuate fibrotic phenotype of PF-bearing mice. Macrophages, which can be polarized to a typically activated phenotype (M1) or alternately activated phenotype (M2), also play an important role in the pathogenesis of pulmonary fibrosis. Previous research has proved that M2 macrophages can produce large quantities of TGF-β1 and platelet-derived growth factor (PDGF), which would induce fibroblast proliferation and differentiation into myofibroblasts, leading to pulmonary fibrosis [69]. A number of studies in our group used cationic liposomes as gene vectors to transport gene therapeutics into the lung of mice with pulmonary fibrosis to reprogram macrophages polarization in the focal area and improve pulmonary fibrosis. Wang et al. [59] generated liposomes loaded with *Mbd2* siRNA (Figure 5A). In vivo imaging showed that liposomes injected through the airway continued to accumulate in the lung for at least 7 days (Figure 5B, Left). Heart, liver, spleen, lung, kidney, and other organs were collected 7 days after intratracheal administration with liposomes. Near-infrared fluorescence (NIRF) signal analysis showed that liposomes were detected only in the lungs, but not in other organs (Figure 5B, Right). The confocal result (Figure 5C) indicated that most of the liposomes (red) accumulated in the fibrotic area and were internalized by macrophages (green). WT mice were injected with *Mbd2* siRNA-loaded liposomes at day 14 and 18 after bleomycin-induced pulmonary fibrosis. Histopathological analysis indicated that *Mbd2* siRNA liposomes could significantly reduce BLM-induced lung injury and fibrosis (Figure 5D). According to the RNA sequencing data, Pan et al. [60] selected Sart1 as the target gene to regulate macrophage polarization. Both in vivo and in vitro studies demonstrated favorable effects of the siRNA-loaded liposomes on attenuating M2 macrophage polarization and improving the PF of BLM-induced mouse model. Interestingly, instead of escaping from the phagocytosis of macrophages, the liposomes employed in the above two studies preferred to be more easily internalized by the macrophage. Attributed to the bounden duty of macrophage is to phagocytize foreign substances, liposomes would mainly accumulate in the macrophages, which makes this strategy quite suitable for the treatment of pulmonary fibrosis by targeting macrophages. In addition, our previous studies demonstrated that sushi-repeat-containing protein X-linked 2 (SRPX2) was overexpressed in the lungs of IPF patients and mice with pulmonary fibrosis. Further functionality studies identified that SRPX2 was involved in a TGFβR1/SMAD3/SRPX2/AP1/SMAD7 positive feedback loop. *Srpx2* siRNA-loaded liposomes were then employed to suppress fibroblast-to-myofibroblast transition for the treatment of pulmonary fibrosis [61]. Our latest study has proven that tartrate-resistant acid phosphatase 5 (ACP5) regulated by TGF -β1 can dephosphorylate p-β-catenin serine 33 and threonine 41, inhibit the degradation of β-catenin, and subsequently enhance β-catenin signaling in the nucleus, leading to the differentiation, proliferation, and migration of fibroblasts and the promotion ofpulmonary fibrosis. After intratracheal injection of *Acp5* siRNA-loaded liposomes in BLM-treated mice, an efficient uptake of liposomes was observed in the fibroblasts of lung lesions, along with decreased levels of fibrotic markers [63]. 

Limitations of cationic lipids include low efficacy and cytotoxicity. Premature release of nucleic acids by interactions with intracellular component and extracellular matrix, clearance by immune cells, and uptake by nonspecific tissues all lead to low efficiency of cationic liposomes. In addition to the ROS production and local inflammation induced by positive charge, the properties, size, and ratio of liposomes to nucleic acids also affect the cytotoxicity of liposomes [12]. These limitations should be taken into account when using liposomes as genetic drug vectors. Neutral lipids, such as phospholipids disaturated phosphatidylcholine (DSPC), dioleoyl phosphatidylethanolamine (DOPE), and membrane component cholesterol, act as “helper lipids” to further increase the interaction between liposomes and endosomal membrane, which can promote siRNA release [12]. Transfection agents such as Lipofectin (DOTMA/DOPE) are based on cationic liposomes used for siRNA transfection [70]. To reduce nonspecific uptake, liposomes are also designed to target specific cells. Otsuka et al. [54] modified the liposome with vitamin A to target the myofibroblast, leading to a specific silence of heat shock protein 47 (HSP47) in pulmonary myofibroblasts, resulting in myofibroblast apoptosis and improving pulmonary fibrosis.

Some other lipid-based gene vectors, such as solid lipid nanoparticles (SLNs) and exosomes, have also been applied in lung delivery. SLNs are generally spherical in the submicron range and composed of solid lipids and surfactant [66]. Compared with liposomes, SLNs exhibit better stability and lower toxicity, as well as controlled release of drugs, but reduced encapsulation efficiency [71]. Sung et al. [72] produced SLNs formulated by tricaprin (TC), DC-Chol, DOPE, and Tween 80 through the melt homogenization method to load pp53–EGFP–plasmid DNA. The transfection efficiency of SLNs was proved to be higher than that of Lipofectin. After the transfection of *p53* plasmid mediated by SLNs into the lung cancer cells (H1299 cells), an overexpression of P53 and restored apoptotic pathway was observed. However, SLNs are not widely used in the therapy of pulmonary diseases, which may be attributed to the fact that the airway administration route requires SLNs to be aerosolized to dry powders, increasing the difficulty and cost of the preparation. Recently, Wang et al. [73] made an aerosolizable siRNA-encapsulated SLNs comprised of *TNF-α* siRNA, lecithin, cholesterol, and a lipid-polyethylene glycol conjugate prepared by thin-film freeze-drying (TFFD). Subsequently, they assessed the transfection efficiency of the dry powder in J774A1 cell line and diffusion efficiency in simulated mucus layer. The particle size, polydispersity index, and Zeta potential of SLNs were retained after TFFD and reconstruction. This provides a potential method for the application of SLNs in pulmonary-drug delivery. 

Exosomes (EXOs), as natural lipid delivery carriers, are endogenous extracellular small vesicles with a diameter of 40–100 nm secreted by various cells and are composed of endogenously synthesized lipid, protein and RNA [74,75]. Exosomes can be internalized by fusing with target cells, activating target cell surface receptors or endocytosis [76]. Compared with liposomes, exosomes exhibit preferable targeted delivery ability as well as little immune response [77]. Synthetic siRNA is usually encapsulated in exosomes by electroporation [76]. However, the loading efficiency of electroporation is quite low [78]. Jeong et al. [79] used human cell-derived exosomes as delivery vehicles to load miRNA-497. This artificially-reconstructed exosome can effectively suppress the neovascularization of endothelial cells and the migration of tumor cells. As accumulating evidences suggested that EXOs were involved in the pathogenesis of lung diseases, Zhang et al. [80] investigated whether inhaled EXOs could be effective delivery vectors to regulate pulmonary immune responses. Firstly, they identified that lung macrophages would efficiently take in intratracheally-instilled serum-derived EXOs in vivo by labeling EXOs with PKH26. Next, they constructed serum-derived EXOs containing *Myd88* siRNA, a well-known adaptor involved in innate immunity, and administrated LPS-treated mice with the exosome complex by intratracheal injection. Downregulation of Myd88 and inflammatory cell counts in BALF, as well as less cellular infiltration in lung tissue through H&E staining, indicated that serum-derived EXOs could successfully deliver small RNA molecules to lung macrophages, which might contribute to future gene therapy for lung diseases. Although EXOs have not been reported as gene vectors for the treatment of pulmonary fibrosis yet, their enhanced biocompatibility, high cellular uptake, reduced toxicity, low immunogenicity, and endosomal escape efficiency also make exosomes potential non-viral gene therapy vectors.

### 4.2. Polymer-Based Gene Vectors

Cationic polymers are widely used in gene therapy due to their extraordinary chemical diversity and functionalization potential. Compared with liposomes, ease of synthesis and lower immunogenicity make polymer-based gene vectors attractive for nucleic acid delivery [81]. 

Polycations, such as polyethylenimine (PEI), chitosan, and dendrimer, contain cationic amine (N) groups that can interact electrostatically with anionic phosphate (P) groups of nucleic acids to form polyplexes (Figure 6). The ratio of amino group to phosphorus group (N/P) directly affects the surface potential, structure, and size of the particles. The multi-cation and high molecular weight (MW) of cationic polymers can improve the transfection efficiency, but the high positive charge may also cause cytotoxicity [82]. The improved transfection efficiency can be explained as a large amount of positive charges within one molecular can tightly condense negatively charged nucleic acid. Additionally, polymers with abundant cations prefer to be efficiently internalized by cells through charge-mediated interactions [82].Cationic polymers help gene therapy drugs escape from lysosomes through the “proton sponge” effect. That is, the cationic polymers enter the target cell through endocytosis to form endosomes, which fuse with the lysosomes. The unsaturated amino group on the cationic polymers chelate the proton provided by the proton pump, resulting in the retention of chloride ions and water molecules in the lysosome, causing the swelling and rupture of the lysosomes, accelerating the release of gene therapy drugs from the endosomes to the cytoplasm [83].

PEI is the most commonly used polymer in gene delivery. PEI can attach to airway epithelial cells and protects the transferred nucleic acid from nuclease degradation [84,85,86]. PEI polyplexes exhibit excellent endosome escaping ability and satisfactory transfection efficiency due to their good gene condensation ability and high buffering capacity. Przybyszewska et al. [51] cloned the soluble receptor I for TNF-a (psTNFR-I) encoding gene into the pcDNA3.1 plasmid to treat radiation-induced PF mice. Compared with naked plasmid, the plasmid/PEI complexes dramatically promoted the incorporating efficacy of psTNFR-I. Moreover, the in vivo studies demonstrated that the plasmid/PEI complexes significantly reduced the pulmonary collagen deposition, alveolar wall thickness, and other histological signs of fibrosis in the mice suffering irradiation-induced pulmonary fibrosis. PEI with specific targeting function can be prepared by directly modifying the PEI main chain with specific ligands. Ding et al. [57] modified PEI with a monocyclam-based CXCR4 antagonist and loaded siRNA to silence plasminogen activator inhibitor-1 (PAI-1), which was a key modulator of ECM production during the pathogenesis of PF. Biodistribution of PEI-C22/siRNA polyplexes was tested by whole-body fluorescent imaging and ex vivo analysis of fluorescence distribution in major organs after intratracheal administration. Whole-body imaging showed that most of the signals were concentrated in the lung, and the retention rate of fluorescence signals could reach to 42% even after 24 h. A suppression of PAI-1 and collagen deposition were observed in the PF-bearing mice after the intratracheal administration of PEI-C22/siPAI-1 polyplexes. Although above studies have corroborated the potential of PEI to be used in the gene therapy of PF, PEI’s cytotoxicity caused by its self-contained intense positive charge limits its application in clinical settings [68]. 

Poly (D, L-lactic-co-glycolic acid) (PLGA), a hydrophobic polymer, has attracted much attention due to its slow release of the payload [87,88]. Modifying cationic polymers or lipids with PLGA is reported to improve their transfection and intracellular transport efficiency. Kolte et al. [89] identified composite nanoparticles of PLGA and PEI could effectively deliver the pDNA to the lung. They prepared NPs with different weight ratios (0–12.5% *w*/*w*) of PLGA/PEI and characterized size, morphology, surface charge, pDNA loading, and in vitro release. The NPs with 10% *w*/*w* PEI exhibited the highest efficacy but unacceptable cytotoxicity. PEG was applied to modify the nanoparticles to reduce toxicity, improve the diffusion of the complexes through the mucus barrier, and prevent the nonspecific uptake by pulmonary macrophages. After being combined with lactose carrier particles, the nanoparticles were lyophilized to dry powder inhaler, serving as a local delivery system of pDNA to lung tissue. In addition to PEGylations, fluorinations are also chosen to modify polycations to achieve better intracellular transport and endosomal escape [90]. Wang et al. [58] reported the fluorinated polymeric CXCR4 inhibitors (PAMD) polyplexes as *PAI-1* siRNA vectors were suitable for pulmonary delivery. 

Chitosan is a natural cationic polysaccharide comprising (1→4) linked 2 amino-2-deoxy-β-d-glucan, which possess favorable biodegradability, biocompatibility, and low toxicity [91]. The excellent abilities of muco-adhesion and muco-penetration make chitosan a potential lung delivery vector [92]. Chitosan can effectively bind with gene drugs, partially protecting nucleic acid from nuclease degradation and facilitating the transport of gene drugs from the cytoplasm to the nucleus by escaping endosome and lysosomal systems [93]. Chitosan can be used as a carrier alone or as a surface modification to improve the transport efficiency of genetic drugs. Nielsen et al. [94] used chitosan as a carrier to transport *EGFP* siRNA and modified the nanoparticles through a nebulizing catheter to convert them into aerosols. After the nanoparticles were injected into the mouse airway through non-invasive endotracheal insertion, the expression of EGFP was decreased in the bronchial epithelium. Ihara et al. [95] prepared dry powdery *EGFP* siRNA–chitosan complexes. Intratracheal injection of the complexes to EGFP transgenic mice led a decreased EGFP expression in the bronchi, bronchioles, and alveolar walls. Gaspar et al. [93] modified SLNs with chitosan as a surface charge modifier. This modification not only increased the interaction between SLNs and negatively charged plasmids, but also enhanced the ability of these particles to be endocytosed by the Calu-3 and A549 cells. Although chitosan has not been applied as a gene vector for the treatment of pulmonary fibrosis, the inherent antifibrotic capability of chitosan makes it quite conspicuous. Kim et al. reported that oral administration of chitosan attenuated bleomycin-induced pulmonary fibrosis in rats by reducing TGF-β1 and IFN-γ levels [96], which makes chitosan a promising gene vector for the treatment of pulmonary fibrosis.

Dendrimer is a unique type of hyperdendritic macromolecule with sizes ranging from 1 to 20 nm, consisting of multi-layered monomer units radiating outwards from the center, and each complete grafting cycle is called a generation [97]. Poly(amidoamine) (PAMAM) dendrimers are the most commonly used dendrimers, attributed to their highly monodispersed, hyperbranched structureand easily functionalized surface groups [98]. Bohr et al. [99] investigated the delivery efficiency of the generation 3 PAMAM dendrimer to transport *TNF-α* siRNA into the lung. Compared with non-complexed siRNA, PAMAM dendritic molecular-siRNA complex (dendriplexes) showed higher levels of cell uptake and TNF-α silencing in the RAW264.7 macrophages. In mice with LPS-induced lung inflammation, the dendriplexes effectively inhibited TNF-α expression in bronchoalveolar lavage fluid and ameliorated acute lung inflammation in mice via pulmonary administration. As reported, a dendrimer-based delivery system is also preferable to be transported to epithelia cells by a portable oral inhalation device. Cont et al. [100] employed a four-generation PAMAM to load siRNA. Then, the dendriplexes were dispersed in mannitol. With the assistance of portable oral inhalation devices, siRNA–*G4NH2* dendriplexes could be efficiently transformed to aerosol, which was quite conducive to deep lung deposition, with respirable fractions of up to 77%. During the process, the bioactivity of siRNA (gene silencing) remained integrated even after the particle preparation processes or long-term exposures to the propellant hydrofluoroalkane (HFA), which is necessary for reconstructing the gene silencing function after delivery to deep lungs. Polymeric micelles are self-assembled nanostructures (10 to 200 nm), consisting of amphiphilic copolymers, which possess good biocompatibility, degradability, and easy modification [101]. However, polymer micelles are usually unstable on the dynamics. To improve the stability of polymer micellar nanomaterials, specific sites of the copolymers can be cross-linked [102]. In addition, the low loading efficiency and the unsatisfactory transmembrane ability also limit their clinical applications [103]. Sung et al. [52] developed a noncovalently post-PEGylated micelle composed of poly(dimethylamino)ethylmethacrylate (PDMAEMA) and its copolymer with poly(α-methylether-ω-methacrylate-ethyleneglycol) [PMAPEG]. After encapsulating *connective tissue growth factor (CTGF)* siRNA, the micelles exhibited excellent antifibrotic effects including reduced collagen deposition, attenuated inflammatory cytokines production, as well as less cytotoxicity compared with PEI. Another study was conducted to inhibit the lung-resident mesenchymal stem cells differentiating to the myofibroblast during the pathogenesis of pulmonary fibrosis. Ji et al. [62] established a target lung-resident mesenchymal stem cells’ (LR-MSCs) micelle by modifying the copolymer PEG-PEI with an anti-stem-cell antigen-1 antibody fragment (Fab′) (Figure 7A). After packaging dual siRNAs, si*RUNX1*, and si*Gli1*, the micelle was administrated to the BLM-induced mice via i.v. injection. The in vivo biodistribution study demonstrated a specific accumulation of micelle in the lung (Figure 7B–E). With the treatment of the micelle, PF-bearing mice exhibited an attenuated disease phenotype and a prolonged overall survival. 

### 4.3. Protein and Peptide-Based Gene Vectors

Protein-based gene vectors, especially those in human bodies, such as serum albumin, transferrin, protamine, and histone, have good biocompatibility, biodegradability, and high biosafety. Furthermore, they are easily modified by other ligands and combined with other functional molecules. Intravenous injection of cationic vectors is limited by the binding of a large number of negatively charged serum components, which may result in rapid clearance of therapeutic agents from the bloodstream. Based on the transient retention of macroaggregated albumin (MAA) in the lungs, Watanabe et al. [49] used MAA–PEI complex to transfer *human hepatocyte growth factor (hHGF)* plasmid to alveolar septa in bleomycin-induced lung fibrosis mice via intravenous administration. Significant decreases of TNF-α, IL-6, and collagen synthesis were observed in the MAA–PEI complex treated mice. Han et al. [104] obtained cationic bovine serum albumin (CBSA) by surface modification of BSA. CBSA can protect siRNA from degradation and significantly enhance the stability of siRNA in serum. More importantly, by optimizing the cationization degree on the surface of CBSA, high siRNA transfer efficiency can be maintained under the premise of low toxicity of the material. Additionally, its interaction with plasma protein in vivo can be controlled to form micron-level complexes intercepted by pulmonary capillaries, leading to an accumulation of CBSA/siRNA complex in the lung. Thus, CBSA can be selectively enriched in the lungs, significantly improving the targeting and therapeutic efficiency of siRNA drugs (Figure 8). Protamine is also a commonly used natural protein vector, which exhibits efficient DNA binding ability and favorable nuclear localization ability [105]. Fukushige et al. [106] developed hyaluronic acid-coated liposome-protamine-siRNA complexes for pulmonary inhalation by spray lyophilization. A significant gene silencing effect in human lung cancer cells was observed after the treatment of the complexes. CRISPR-Cas9 system is a powerful technology that relies on the Cas9/sgRNA ribonucleoprotein complexes (RNPs) to target and edit DNA [107]. Delivery of Cas9 RNPs requires electroporation or transfection mediated by lipid- or polymer-vectors [107]. Kim et al. [108] established a multifunctional Cas9 fusion protein (Cas9-LMWP) carrying both a nuclear localization sequence and a low molecular weight protamine (LMWP). Cas9-LMWP enabled the direct self-assembly of a Cas9:crRNA:tracrRNA ternary complex (ternary Cas9 RNP) and delivered the ternary Cas9 RNPs into the recipient cells. Like protamine, histones are natural nucleic acid-binding proteins. The ability of histones to condense nucleic acids and their multiple nuclear localization signals made them effective gene vectors [109]. However, studies on histones as gene vectors in the treatment of lung diseases have not yet been found, which may be due to the fact that the histone itself also plays an important role in the pathological process of lung inflammation and fibrosis [110,111].

Peptides are short chains of amino acids composed of no more than 50 amino acid residues, which can be used as gene vectors alone or as a functional component to participate in gene delivery system to achieve efficient gene transfection [112,113]. Cell-penetrating peptides (CPPs) are short peptides that can penetrate biofilms and deliver a variety of bioactive substances into cells. Covalent conjugation and non-covalent complexation are two ways to deliver cargo molecules with peptides. They cross cell membranes through endocytosis and energy-independent pathways (Figure 9), alongwith low cytotoxicity and little immune response [114]. As carriers of negatively charged nucleic acids, positively charged CPPs also protect the gene therapeutics from enzymatic degradation in the airway mucus. Ding et al. [56] evaluated the anti-fibrosis capacity of silenced *SPARC*, *CCR2*, and *SMAD3* by loading siRNA with CADY peptide nanoparticles. The expression levels of Sparc, Ccr2, and Smad3 in the treated group were significantly reduced by intraperitoneal injection of nanoparticles into bleomycin-induced pulmonary fibrosis mice on days 10, 14, and 18. Moreover, ameliorated fibrosis in lungs tissues was observed. Ishiguro et al. [115] used calcium chloride to condense dimerized TAT peptide (dTAT) and plasmid *angiotensin II type 2 receptor (pAT2R)* complexes (dTAT-pAT2R-Ca^2+^). Intratracheal aerosol spray or intravenous injections of the complexes significantly alleviated the acute growth of the lung carcinoma in mice models, which indicated a desirable ability of dTAT for DNA delivery. When used as surface modifiers, CPPs can be incorporated into liposome or polymer systems via covalent binding or electrostatic interactions [116]. Jeong et al. [117] utilized CPPs consisting of arginine with spacer arm to modify chitosan/siRNA nanoparticles and modification with the peptides enhanced uptake of the nanoparticles by the mouse airway epithelial cells. Fusogenic peptides promote the endosomal escape ofvectors by increasing the interaction between vectors and endosomal membranes [118]. Glu-Ala-Leu-Ala (GALA) and Lys-Ala-Leu-Ala (KALA) are pH-sensitive fusogenic peptides that undergo structural changes upon pH changes to facilitate the endosomal release of the vectors. Kenji et al. [119] reported the multifunctional envelope-type nano device (MEND) modified with a GALA peptide (GALA/MEND) as a siRNA vector which effectively targeted the pulmonary endothelium. This provides a good option for the future treatment of pulmonary fibrosis. 

### 4.4. Physical Properties of Non-Viral Gene Vectors

The therapeutic effect of non-viral gene pharmaceuticals is closely related to their physical properties. The size, Zeta potential, and uniformity as well as colloidal stability of the gene vector can be used to predict the efficacy of the agent in clinical settings [121,122]. Among these parameters, the particle size of nanoparticles is particularly important, especially in the case of pulmonary or parenteral drug delivery, which plays a key role in the effective delivery of drug active ingredients [123]. Liposomes with large particle size (>300 nm) lack vascular permeability and cannot pass through the intercellular space of hepatic blood vessels and are easily swallowed by the reticuloendothelial system, leading to a short half-life in vivo [124]. Liposomes with particle size less than 300 nm can escape from the uptake of liver and spleen and thus increase the accumulation of gene therapeutics in target sites [124]. In addition, the different degrees of fusion and aggregation of carriers also affect the storage time of carriers [125]. The surface potential of nanoparticles is another important physicochemical parameter because it determines the strength of the interactions within the particles, the adsorption of counter-ions, and thus the stability of the particles [125]. This parameter is usually expressed as the Zeta potential, which describes the charge distribution of bare particles associated with the diffusion layer. The higher the absolute value of Zeta potential is, the greater the charge on the surface of the gene carrier is. With the increase in electrostatic repulsion caused by the double electric layer, the vectors have to overcome huge amounts of energy to aggregate, improving the stability of the vectors. Furthermore, the level of Zeta potential also indicates its ability to compress nucleic acid. Generally speaking, the vectors featured with the high positive charge can compress more nucleic acid. However, at the same time, the greater cell toxicity will be accompanied. Therefore, the Zeta potential of the nanoparticles should be controlled in an appropriate range [121]. In general, when the absolute value of Zeta potential is less than 30 mV, the charged particles are unstable and easy to aggregate. When the absolute value of Zeta potential is greater than 30 mV, it has better electrostatic stability [125].

## 5. Conclusions and Prospects

Due to the lack of effective pharmaceutics, the mortality of PF is still stubbornly high. Nowadays, the development of gene therapy brings new hope for the treatment of PF. However, the transfection efficiency and the stability of naked nucleic acids are always unsatisfactory, which impede their clinical applications. Thus, an appropriate gene carrier is urgently needed for the gene therapy of PF.

Up to now, there are no commercial products or even clinical trials using non-viral gene strategies to treat pulmonary fibrosis. In recent years, due to the rapid spread of the COVID-19 epidemic, Pfizer BioNTech and Moderna have developed mRNA-based lipid nanoparticles for COVID-19 vaccines, which have received emergency authorization from the FDA and have achieved 95% and 94.1% efficacy against COVID-19, respectively [126,127] (Table 2). These indicate the prospects of gene therapy with non-viral vectors for pulmonary disease, inspiring us to move the concept of curing PF with non-viral gene therapy from bench to bedside.

With the in-depth understanding of the pathogenesis of pulmonary fibrosis and the rapid development of materials science, non-viral vectors have emerged as promising tools for the gene therapy of PF. Compared with viral vectors, non-viral vectors have shown lower immunogenicity, higher loading efficiency, easier synthesis, and lower cost, making them more appropriate for widespread clinical use. Due to the special physiological function and anatomical structure of the lung, the non-viral vectors can be administrated through the airway, which is more convenient than systemic administration. Moreover, inhalation administration of non-viral vectors exhibits better treatment compliance for patients, attributed to the dramatically elevated accumulating amount of drugs in the lung and minimized the side effects.

Although non-viral vectors accelerate the pace of gene therapy in PF treatment, there are still some problems demanding prompt solutions. Compared to viral vectors, non-viral vectors exhibited lower delivery efficiency, which is the main reason why few of these vectors have been clinically developed to date. Up to now, only liposomes have been approved as non-viral gene therapeutic carriers for clinical applications, while the safety of the other materials remains to be understood. Another critical issue is that most of the current studies are basically carried out on rodent models. Due to the huge differences in airways between rodents and humans, the therapeutic strategies suitable for mice might not be fit for the patients. Additionally, PF bearing animal models cannot completely reproduce the pathologies and pathogenesis of IPF patients. Lastly, the majority of studies are conducted in the early stage of PF, while pulmonary fibrosis, once detected, is usually in the middle or late stages in clinic. Therefore, the PF models need to be optimized for the evaluation of non-viral gene vectors. Nevertheless, the non-viral gene therapy remains the most promising strategy for the treatment of PF by far, which deserves to be developed.

## Figures and Tables

**Figure 1 pharmaceutics-14-00813-f001:**
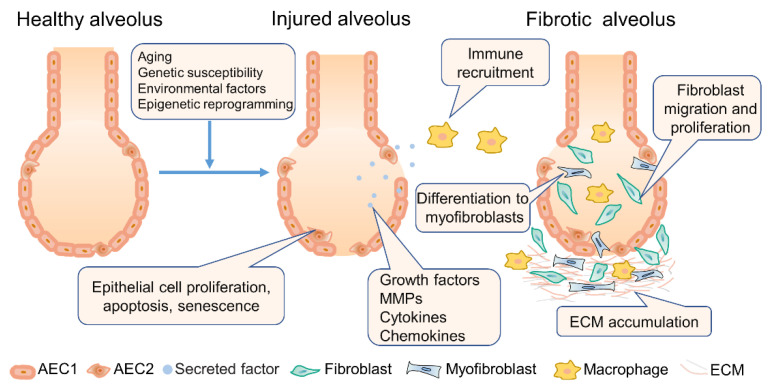
Pathophysiology of IPF.

**Figure 2 pharmaceutics-14-00813-f002:**
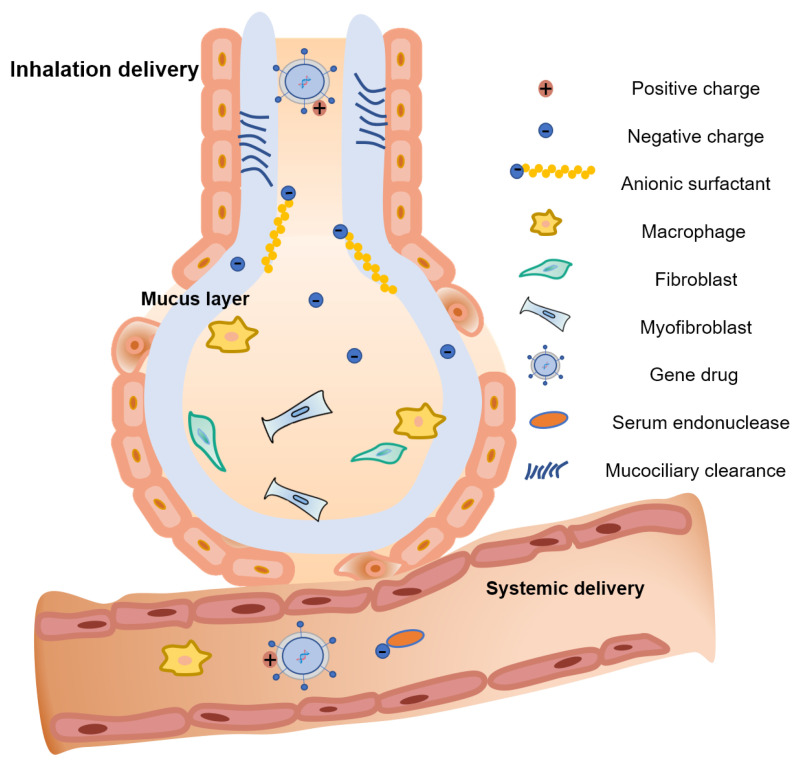
Barriers for the gene therapy of pulmonary fibrosis.

**Figure 3 pharmaceutics-14-00813-f003:**
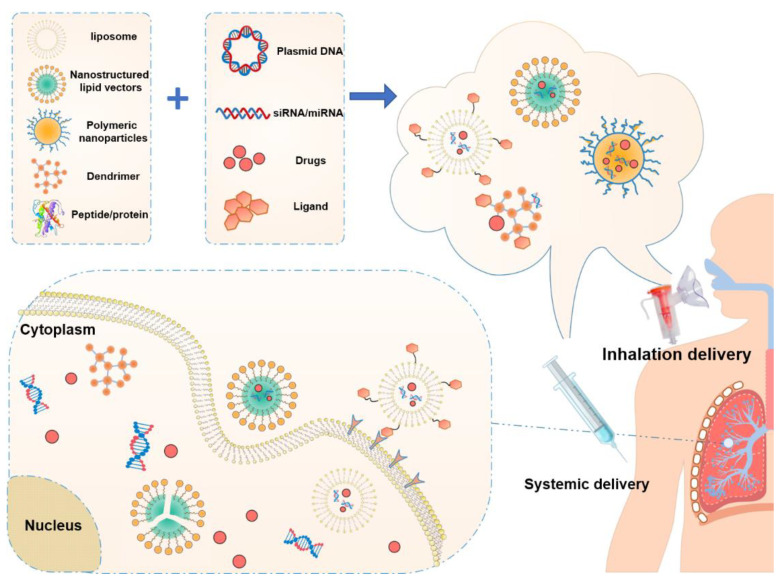
Pulmonary and systemic delivery of gene drugs to the lung.

**Figure 4 pharmaceutics-14-00813-f004:**
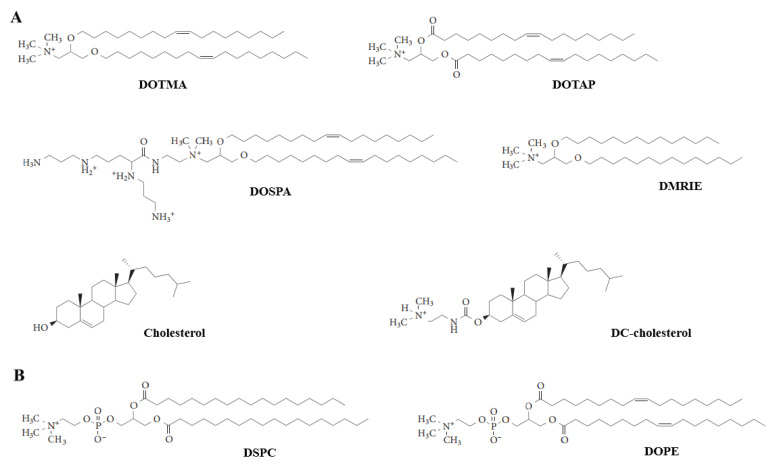
Chemical structures of lipid-based vectors [12]. (**A**) Cationic lipids and (**B**) neutral lipids.

**Figure 5 pharmaceutics-14-00813-f005:**
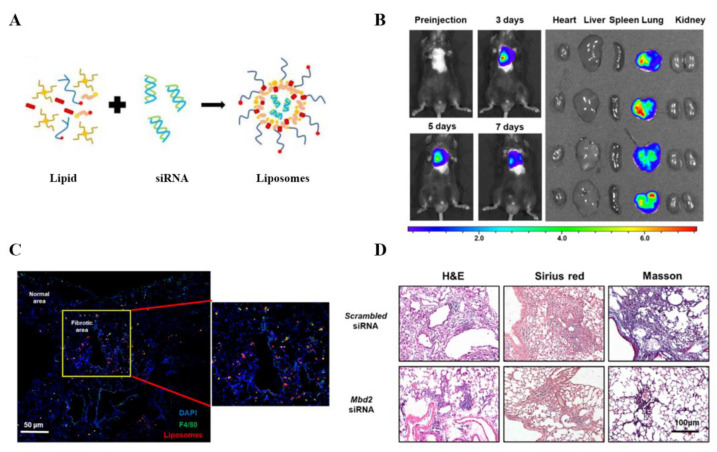
*Mbd2* siRNA-loaded liposomes protected mice from BLM-induced pulmonary fibrosis [59]. (**A**) Schematic diagram of *Mbd2* siRNA-loaded liposomes preparation. (**B**) Representative in vivo images of the mouseintratracheally administrated with DiR-labeled liposomes (**Left**) and ex vivo images of major organs from mice (**Right**). (**C**) Thebiodistribution of liposomes in lungs from BLM-induced mice. (**D**) Representative results for H&E, Sirius red, and Masson staining indicated intratracheal administration of *Mbd2* siRNA–loaded liposomes provided protection for mice against BLM induced pulmonary fibrosis.

**Figure 6 pharmaceutics-14-00813-f006:**
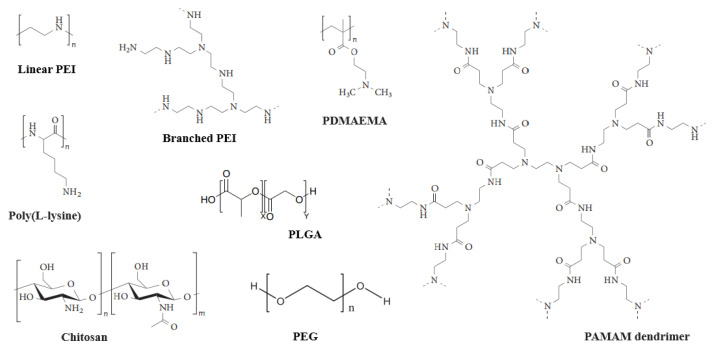
Chemical structures of polymer-based gene vectors [12].

**Figure 7 pharmaceutics-14-00813-f007:**
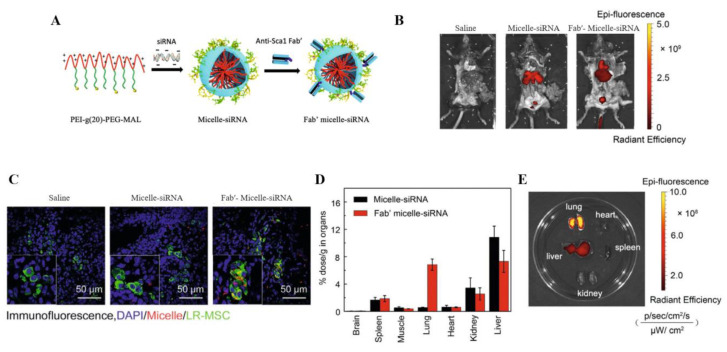
Targeted siRNA delivery by functionalizing the micelle surface with an anti-stem-cell antigen-1 antibody fragment (Fab’) for pulmonary fibrosis therapy [62]. (**A**) Schematic diagram of the siRNA-loaded polymeric micelles targeting lung mesenchymal stem cells. (**B**,**C**) Biodistribution of the delivered micelles. (**D**) The biodistribution of intravenously administrated micelle–siRNA and anti-Sca1 Fab′-conjugated Micelle-siRNA in organs and (**E**) images of the organs of a mouse administrated with Fab′–Micelle–siRNA.

**Figure 8 pharmaceutics-14-00813-f008:**
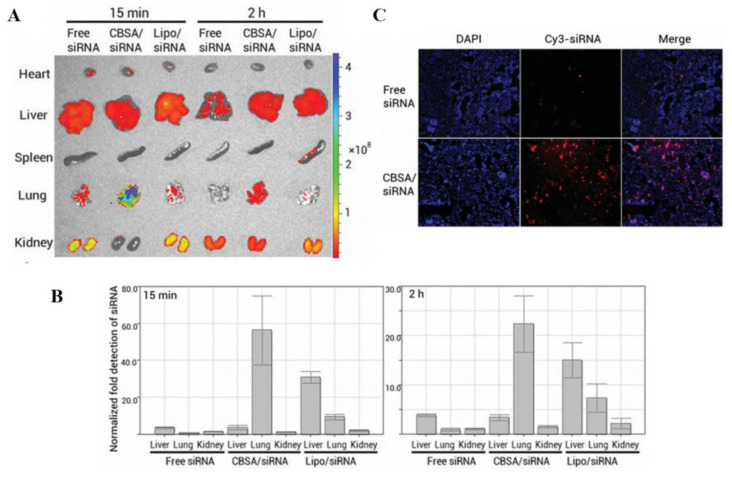
CBSA serves as a siRNA delivery vector for treating lung disease [104]. (**A**) Representative ex vivo fluorescence image of major organs from mice sacrificed at 15 min or 2 h after i.v. injection of Cy3-labeled siRNA alone or complexed with CBSA or Lipofectamine 2000. (**B**) RT-PCR analysis of the silencing efficiency of siRNA in major organs. (**C**) Representative images of lung tissue section.

**Figure 9 pharmaceutics-14-00813-f009:**
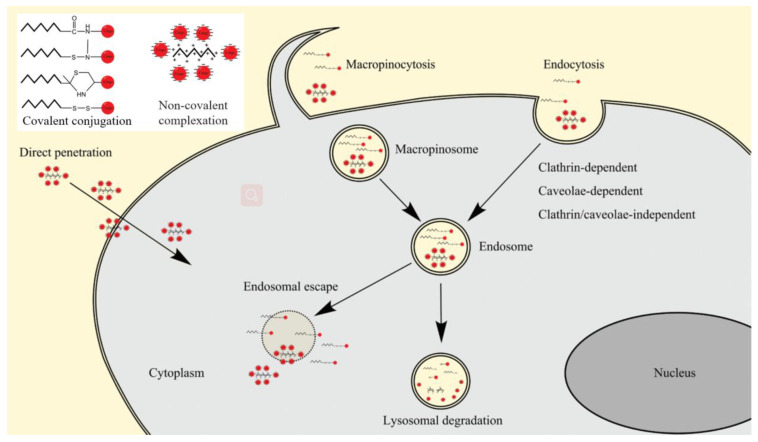
CPPs uptake involved pathways [120].

**Table 1 pharmaceutics-14-00813-t001:** Publications on non-viral gene vectors for pulmonary fibrosis.

Vector	Composition	Target Gene	Nucleotide	Expression	Species	Route	Model	Year	Ref.
Liposomes	DOTMA, DOPE	SOD2	plasmid	Up	mice	i.t.	bleomycin	1999	[48]
Polycations	MAA-PEI	HGF	plasmid	Up	mice	i.v.	bleomycin	2005	[49]
Liposomes	DharmaFECT™ 1	SPARC	siRNA	down	mice	i.t.	radiation	2010	[50]
Polycation	PEI	psTNFR-I	plasmid	Up	mice	i.m.	bleomycin	2011	[51]
Polycations	PMAPEG, PDMAEMA	CTGF	siRNA	down	mice	i.t.	bleomycin	2013	[52]
Polycations	DODMA, DSPC, PEG	AR or CTGF	siRNA	down	mice	i.t.	bleomycin	2016	[53]
Liposomes	vitamin A, DC-6-14	HSP47	siRNA	down	rats	i.t.	bleomycin	2017	[54]
Liposomes	NLCs-PGE2	MMP3, CCL12, HIF1A	siRNA	down	mice	i.t.	bleomycin	2017	[55]
Peptides	CADY peptide	SPARC, CCR2, SMAD3	siRNA	down	mice	i.p.	bleomycin	2018	[56]
Polycations	PEI-C22	PAI-1	siRNA	down	mice	i.t.	bleomycin	2019	[57]
Polycations	F-PAMD	PAI-1	siRNA	down	mice	i.t.	bleomycin	2019	[58]
Liposomes	C12-200, mPEG-DMG	MBD2	siRNA	down	mice	i.t.	bleomycin	2021	[59]
Liposomes	C12-200, mPEG-DMG	SART1	siRNA	down	mice	i.t.	bleomycin	2021	[60]
Liposomes	C12-200, mPEG-DMG	SRPX2	siRNA	down	mice	i.t.	bleomycin	2021	[61]
Polycations	PEI-g-PEG-Mal	RUNX1 or Gli1	siRNA	down	mice	i.v.	bleomycin	2021	[62]
Liposomes	C12-200, mPEG-DMG	ACP5	siRNA	down	mice	i.t.	bleomycin	2022	[63]
Polycations	PFC nanoemulsions	STAT3	siRNA	down	mice	i.t.	bleomycin	2022	[64]

**Table 2 pharmaceutics-14-00813-t002:** Non-viral gene vectors for COVID-19 vaccines.

Nucleic Acid	Generic Name	Vector	Composition	Company	Route	Clinical Trial	Status
mRNA	BNT162b2	LNPs	ALC-3015, ALC-0159, DPSC	Pfizer BioNTech	i.m.	NCT04283461	Active
mRNA	mRNA-1273	LNPs	SM-102, PEG2000-DMG, DSPC	Moderna	i.m.	NCT04470427	Active

## Data Availability

Not applicable.

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
