# Peer review of "Treating Pulmonary Fibrosis with Non-Viral Gene Therapy: From Bench to Bedside"

_pharmaceutics, 2022, doi:10.3390/pharmaceutics14040813_

Round 1
Reviewer 1 Report
The review article done by Huang et al. is a very interesting one and approaches a hot topic.
The paper is clear, well written and the conclusions are supported by the results.
My only minor comments is related to the lack of some figure or schemes from the second part of the paper. I would have preferred to see at least one result for each type on non-viral vector.
Author Response
The review article done by Huang et al. is a very interesting one and approaches a hot topic. The paper is clear, well written and the conclusions are supported by the results. My only minor comment is related to the lack of some figure or schemes from the second part of the paper. I would have preferred to see at least one result for each type on non-viral vector.
Responses: We really appreciate this suggestion, which would improve our manuscript a lot. We added 3 figures (Figure 5, 7 and 8) for the second part of the paper.
Reviewer 2 Report
Huang and co-workers reviewed non-viral delivery systems for potential treatment of pulmonary fibrosis. The authors focused on nanovectors based on lipids, polymers and proteins/peptides. The manuscript is written in a systematic and clear way with some original figures. However, chemical structures of new or lesser known lipids and polymers should be added to the manuscript, as well as figures for some research studies discussed in order to facilitate/improve the understanding of the studies. Other concerns should also be addressed:
- In Table 1, more detail concerning type of vector should be added, such as, name(s) of polymers/lipids;
- In line 242 “siRNA” should be modified to “nucleic acids”;
- In line 300, “Polymers” should be modified to “Cationic polymers”;
- In lines 307/309, authors should add the explanation for this phenomenon;
- In line 348 authors should correct “choosed” to “chose”;
- In lines 383/384 the authors should describe and discuss the results obtained in this research study;
- A section only devoted to future perspectives must be added to the manuscript. It would be desirable to dedicate a section to a detailed commercial potential (products in the market, patents and actual clinical trials);
- All acronyms should be described when appear the first time in the text and the acronym must be placed in parentheses after that;
- Verb contraction should be avoid, as well as some expressions like "etc";
- The authors should carefully revise the manuscript to correct typos like “polycatatations”, “antifibiotic”, “furthermoer”, “issuse” among others.
There is a recent review paper on this topic (https://doi.org/10.1016/j.omtm.2021.01.003 ) and for this reason I am afraid regarding the novelty of this paper.
Author Response
- Huang and co-workers reviewed non-viral delivery systems for potential treatment of pulmonary fibrosis. The authors focused on nanovectors based on lipids, polymers and proteins/peptides. The manuscript is written in a systematic and clear way with some original figures. However, chemical structures of new or lesser known lipids and polymers should be added to the manuscript, as well as figures for some research studies discussed in order to facilitate/improve the understanding of the studies.
Responses: We appreciate this constructive suggestion, and add 3 figures for chemical structures of the vectors (Figure 4, 6 and 9) and 3 figures for some research studies (Figure 5, 7 and 8).
Other concerns should also be addressed:
- In Table 1, more detail concerning type of vector should be added, such as, name(s) of polymers/lipids.
Responses: Thanks a lot for your constructive suggestions. We have supplied major components for each vector in Table 1.
- In line 242 “siRNA” should be modified to “nucleic acids”.
Responses: We changed “siRNA” to “nucleic acids” in the revised manuscript.
- In line 300, “Polymers” should be modified to “Cationic polymers”.
Responses: We modified “Polymers” to “Cationic polymers” in the revised version.
- In lines 307/309, authors should add the explanation for this phenomenon.
Responses: As suggested, we add the explanation for the phenomenon in the revised manuscript.
- In line 348 authors should correct “choosed” to “chose”.
Responses: We modified “choosed” to “chose” in the revised version.
- In lines 383/384 the authors should describe and discuss the results obtained in this research study;
Responses: We describe and discuss the results obtained in this research study.
- A section only devoted to future perspectives must be added to the manuscript. It would be desirable to dedicate a section to a detailed commercial potential (products in the market, patents and actual clinical trials);
Responses: We added a section to present actual clinical trials in non-viral gene vectors in conclusions and prospects.
- All acronyms should be described when appear the first time in the text and the acronym must be placed in parentheses after that.
Responses: We described acronyms when appear the first time in the text.
- Verb contraction should be avoid, as well as some expressions like "etc";
Responses: We deleted the verb contraction.
- The authors should carefully revise the manuscript to correct typos like “polycatatations”, “antifibiotic”, “furthermoer”, “issuse” among others.
Responses: We apologized for our mistake and corrected the typos.
- There is a recent review paper on this topic (https://doi.org/10.1016/j.omtm.2021.01.003 ) and for this reason I am afraid regarding the novelty of this paper.
Responses: We really appreciate this friendly reminder and carefully read this article. The authors reviewed publications on the use of gene therapies to treat pulmonary fibrosis in animals, and summarized gene therapies for pulmonary fibrosis in terms of enhancing, silencing or repressing gene expression. They mainly focus on the nucleic acid applied in gene therapy, other than vectors. However, we focused on non-viral gene vectors and discussed the existing non-viral vector-based gene therapies by introducing the classification, properties and other characteristics.
Reviewer 3 Report
The manuscript entitled “Treating pulmonary fibrosis with non-viral gene therapy: from bench to bedside” by Huang et al. reports recent development in gene delivery for the treatment of idiopathic pulmonary fibrosis (IPF) for the carrier such as liposomes, polymers, and proteins/peptides. This review provides concise and up-to-date literature in the introduction for the IPF followed by pathophysiology for the disease. Authors further describe the gene therapeutics for pulmonary diseases and the non-viral gene delivery agents. Table 1 was provided with various non-viral gene vectors. Overall, a good and concise review in the field to recommend for publication after minor suggested revision. Authors are advised to add a table to report the patents and clinical trials for PF gene therapy using non-viral agents. In section 5(conclusions and prospects), the author should add some more pharmaceutic prospects/properties for clinical use of nonviral delivery agents, by comprehend with the example of any clinical agent available for PF.
Author Response
The manuscript entitled “Treating pulmonary fibrosis with non-viral gene therapy: from bench to bedside” by Huang et al. reports recent development in gene delivery for the treatment of idiopathic pulmonary fibrosis (IPF) for the carrier such as liposomes, polymers, and proteins/peptides. This review provides concise and up-to-date literature in the introduction for the IPF followed by pathophysiology for the disease. Authors further describe the gene therapeutics for pulmonary diseases and the non-viral gene delivery agents. Table 1 was provided with various non-viral gene vectors. Overall, a good and concise review in the field to recommend for publication after minor suggested revision. Authors are advised to add a table to report the patents and clinical trials for PF gene therapy using non-viral agents. In section 5 (conclusions and prospects), the author should add some more pharmaceutic prospects/properties for clinical use of nonviral delivery agents, by comprehend with the example of any clinical agent available for PF.
Responses: Many thanks for your suggestion. We carefully searched the Clinicaltrials.gov and found no clinical trials of non-viral gene therapy related to pulmonary fibrosis. At present, clinical studies on non-viral gene therapy for lung diseases are mainly focused on lung cancer, obstructive lung diseases (such as asthma, COPD, cystic fibrosis, etc.) and COVID-19. In section 5 (conclusions and prospects), we added some more pharmaceutic prospects/properties for clinical use of non-viral delivery agents in the revised version.
Reviewer 4 Report
In this study, the authors conduct a literature evaluation of different non-viral nanosystems that could be used in the treatment of IPF. The manuscript is interesting, but the following points need to be addressed before its publication.
Major concerns:
- The introduction should be improved by giving global information on recent IPF cases (if statistics are available) and, if possible, identifying the disease's occurrence within the population (men, women, children, etc.) in a broad sense.
- The types of non-viral vectors that will be researched in this work should be briefly mentioned in the introduction.
- It might be conceivable to encapsulate the two primary medications used to treat IPF (nintedanib and pirfenidone) in liposomes or polymeric nanoparticles, for example, with the goal of reducing drug concentration and thereby reducing undesirable side effects. If that's the case, given that the authors stress the importance of discovering new medications in the introduction, this application would be worth mentioning.
- The authors claim the following on lines 165-167: “Compared with viral vectors, non-viral vectors can encapsulate more nucleic acids. The preparation of non-viral vectors is simple and the production cost is much cheaper than that of viral vectors. In addition, non-viral vectors possess lower immunogenicity than that of viral vectors”. These claims must be backed up by bibliographical references.
- The size of polymeric nanoparticles, for example, has been demonstrated to be critical in determining the encapsulation ability of certain medications or, in this case, RNA; however, the authors only provide size information for hexosomes. What sizes are appropriate for using liposomes or polymeric nanoparticles in this application? Please delve further into this.
- In general, the authors should mention the physicochemical properties of the materials used to develop the polymeric vectors, such as zeta potential, morphology, particle density, and so on, because these properties can influence the encapsulation process of any compound, whether it is a drug, a gene, etc.
Minor concerns:
- A table of abbreviations is required due to the vast number of acronyms used in this work.
Author Response
Major concerns:
- The introduction should be improved by giving global information on recent IPF cases (if statistics are available) and, if possible, identifying the disease's occurrence within the population (men, women, children, etc.) in a broad sense.
Responses: We really appreciate this suggestion and improved the paper by giving global information on recent IPF cases.
- The types of non-viral vectors that will be researched in this work should be briefly mentioned in the introduction.
Responses: We added briefly information about non-viral vectors in the introduction.
- It might be conceivable to encapsulate the two primary medications used to treat IPF (nintedanib and pirfenidone) in liposomes or polymeric nanoparticles, for example, with the goal of reducing drug concentration and thereby reducing undesirable side effects. If that's the case, given that the authors stress the importance of discovering new medications in the introduction, this application would be worth mentioning.
Responses: We added some discuss about nanoparticles loading the drugs in the treatment of pulmonary fibrosis in section 2.
- The authors claim the following on lines 165-167: “Compared with viral vectors, non-viral vectors can encapsulate more nucleic acids. The preparation of non-viral vectors is simple and the production cost is much cheaper than that of viral vectors. In addition, non-viral vectors possess lower immunogenicity than that of viral vectors”. These claims must be backed up by bibliographical references.
Responses: We apologized for our mistake and added references to the text.
- The size of polymeric nanoparticles, for example, has been demonstrated to be critical in determining the encapsulation ability of certain medications or, in this case, RNA; however, the authors only provide size information for hexosomes. What sizes are appropriate for using liposomes or polymeric nanoparticles in this application? Please delve further into this.
Responses: We added the size information about non-viral vectors in the revised manuscript.
- In general, the authors should mention the physicochemical properties of the materials used to develop the polymeric vectors, such as zeta potential, morphology, particle density, and so on, because these properties can influence the encapsulation process of any compound, whether it is a drug, a gene, etc.
Responses: We added a paragraph to introduce physical properties of non-viral vectors (section 4.4) in the revised manuscript.
Minor concerns:
- A table of abbreviations is required due to the vast number of acronyms used in this work.
Responses: We added abbreviations in the revised manuscript.
Round 2
Reviewer 2 Report
The authors accepted the suggestions recommended. The quality of the manuscript was improved.
Reviewer 4 Report
Just a suggestion, perhaps the abbreviations should be placed after the conclusions at the end of the manuscript. A list of abbreviations is more convenient, while I understand that in this case, a list can consume a lot of space.